# Integrated Analytical Tools for Accessing Acridones and Unrelated Phenylacrylamides from *Swinglea glutinosa*

**DOI:** 10.3390/molecules25010153

**Published:** 2019-12-30

**Authors:** Ana Calheiros de Carvalho, Luiza De Camillis Rodrigues, Alany Ingrid Ribeiro, Maria Fátima das Graças Fernandes da Silva, Lívia Soman de Medeiros, Thiago André Moura Veiga

**Affiliations:** 1Programa de Pós-Graduação em Biologia Química, Department of Chemistry, Federal University of São Paulo, Diadema-SP 09972-270, Brazil; carvalho.ac08@gmail.com; 2Department of Chemistry, Federal University of São Paulo, Diadema-SP 09972-270, Brazil; decamillisluiza@gmail.com (L.D.C.R.); livia.soman@unifesp.br (L.S.d.M.); 3Department of Chemistry, Federal University of São Carlos, São Carlos-SP 13565-905, Brazildmfs@ufscar.br (M.F.d.G.F.d.S.)

**Keywords:** *Swinglea glutinosa*, dereplication, acridones, phenylacrylamides

## Abstract

In natural product studies, the purification of metabolites is an important challenge. To accelerate this step, alternatives such as integrated analytical tools should be employed. Based on this, the chemical study of *Swinglea glutinosa* (Rutaceae) was performed using two rapid dereplication strategies: *Target Analysis* (Bruker Daltonics^®^, Bremen, Germany) MS data analysis combined with MS/MS data obtained from the GNPS platform. Through UHPLC-HRMS data, the first approach allowed, from crude fractions, a quick and visual identification of compounds already reported in the *Swinglea* genus. Aside from this, by grouping compounds according to their fragmentation patterns, the second approach enabled the detection of eight molecular families, which presented matches for acridonic alkaloids, phenylacrylamides, and flavonoids. Unrelated compounds for *S. glutinosa* have been isolated and characterized by NMR experiments, Lansamide I, Lansiumamide B, Lansiumamide C, and *N*-(2-phenylethyl)cinnamamide.

## 1. Introduction

Currently, a combination of hyphenated techniques (i.e., two or more analytical techniques) may increase the efficiency and speed of analysis, being useful tools to determine unknown natural products. Recent methodologies developed to discover new metabolites include molecular dereplication, which is defined as the analysis of a natural product, fraction, or crude extract without previous purification steps. Usually, this is done based on spectroscopic, structural, or biological activity, using data comparisons obtained from “in-house” and/or commercial databases [1].

In this sense, one of the most employed approaches is the Global Natural Products Social Molecular Networking (GNPS), which consists of a database that analyzes mass spectrometry data and compares it with previously registered data to establish the molecular networking maps. GNPS has been created to improve and accelerate the discovery of natural products, allowing the identification of substances not yet reported [2].

Another tool recently developed to distinguish known and unknown secondary metabolites is HRMS data processing through Target Analysis software (Bruker Daltonics^®^) [3]. This screening method interacts with previously known compound databases by an internal application (Excel spreadsheet) that generates searching lists, which indicate reported detected compounds. This enables accelerated and efficient identification of known compounds, saving time for isolating unknown compounds or bioactive substances. This strategy was developed by Klitgaard et al. (2013) [3].

Based on the advantages of the application of modern strategies, this work aims to explore the chemical profile of *Swinglea glutinosa*, a species from the Rutaceae family, which belongs to a monotypic genus, according to Engler (1931) [4]. It is a plant from the Philippines, but is already widespread throughout the world including Latin America, especially Colombia and Brazil. Biosynthetically, it is characterized by the presence of alkaloids, especially acridones [5] and benzoyltyramines [6].

Some reports have shown that acridones present antiparasitic activity against *Plasmodium falciparum* and *Trypanosoma brucei rhodesiense*, which are responsible for transmitting malaria and sleeping sickness, respectively. Acridone 5-hydroxynoracronycine (**6**), among those tested, was the most active against *T. b. rhodesiense* (IC^T^_50_ 1.0 µM). On the other hand, glycocitrine-IV (**5**), was more active (IC^P^_50_ 0.3 µM) against *P. falciparum* [7]. The acridones also presented an effect on cathepsin V, an enzyme that degrades random proteins in the lysosome, which is associated with some diseases, the progression of tumors, muscular dystrophy, Alzheimer’s disease, rheumatoid arthritis, and osteoporosis. Among the tested compounds, citibrasine (**4**) was the most potent inhibitor, with an IC_50_ value of 1.2 μM [8].

Beyond these effects, we can find reports on the potential of this class of compounds on photosynthesis inhibition. Citrusinine-I (**1**), glycocitrine-IV (**5**), 1,3,5-trihydroxy-10-methyl-2,8-bis(3-methylbut-2-en-1-yl)-9(10*H*)-acridinone, (2*R*)-2-*tert*-butyl-3,10-dihydro-4,9-dihydroxy-11-methoxy-10-methylfuro-[3,2-b]acridin-5(2*H*)-one, and (3*R*)-2,3,4,7-tetrahydro-3,5,8-trihydroxy-6-methoxy-2,2,7-trimethyl-12*H*-pyrano [2,3-a]acridin-12-one affect photosynthesis through different mechanisms of action [5]. We also can find reports on anticancer activity, for instance, compound 1,3-dimethoxy-10-methylacridone, which presented cytotoxic effects with IC_50_ values from 3.38 μM (toward MDA-MB-231-BCRP cells) to 58.10 μM (toward leukemia CEM/ADR5000 cells) [9].

Given the reports and the biological activities associated with compounds isolated from *Swinglea glutinosa*, we have decided to continue [5] our search for compounds still undiscovered in the plant. Thus, the selected modern analytical tools have been very useful for conducting this work, which led us to isolate and characterize substances of interest, in this case, unrelated phenylacrylamides to the *Swinglea* genus.

## 2. Results and Discussion

Before starting the chemical fractionation of *S. glutinosa* extracts, to detail the chemical profile of the plant, a literature review (including the use of the Dictionary of Natural Products) of all compounds previously reported for the *Swinglea* genus was performed. Thus, an “in-house” database was created by feeding an Excel spreadsheet containing the molecular formula and the name of all cataloged compounds. In total, 27 compounds were cataloged, belonging to the acridone and benzoyltyramine classes.

Among the fractions obtained from the ethanolic extract fractionation of *S. glutinosa*, the hexane stem and hexane leaf fractions were analyzed through the dereplication approaches. Thus, it was possible to observe on the chromatogram of the hexane stem fraction that many detected compounds corresponded to compounds listed in the “in-house” database, most of them belonging to the acridonic alkaloid, benzoyltyramine, and phenylacrylamide classes (Figure 1A and Figure 2; Table 1). The numbers indicated on the chromatograms (Figure 1A,B) correspond to the molecular formulas for the compounds present in the “in-house” database. These compounds are shown in Figure 2 and Table 1.

On the other hand, from the analysis of the hexane leaf fraction (Figure 1B), we observed that its major compounds did not correspond to the cataloged metabolites in our database. To find out which classes of compounds were present in the fraction as well as in the other fractionated amounts, we decided to use a complementary dereplication strategy: the free website GNPS.

Currently, the use of molecular networking is a powerful analytical tool for metabolic mapping by molecular fragmentation data through tandem mass spectrometry [2]. This makes it possible to represent and to group a set of spectral data based on the fragmentation similarity (MS/MS spectra) of compounds present in one or more target samples. Directly, such grouping suggests a structural similarity between compounds, thus facilitating the detection of biosynthetic analogues [10]. Therefore, through the analysis of the obtained molecular families from the extracts of *S. glutinosa* (Figure 3 and Appendix A), it was possible to visualize the establishment of eight predominant clusters.

The orange and green colors represented in the nodes (Figure 3) illustrate the presence of the described precursor ions found in the extracts from the stems and leaves of the plant, respectively. It is important to highlight that the indicated proportions should not be associated with the amounts of metabolite detected in each extract. The observed differences correspond to the number of spectral counts recorded for each ion, according to the program processing standardization.

The molecular family I indicates the detection of seven metabolites belonging to the *N*-benzoyltyramine class, a known group of compounds found in the *Swinglea* genus [6]. However, all seven biosynthetic congeners have not been described for *S. glutinosa* yet. Given this, we decided to isolate the compounds represented by *m*/*z* 264.104, 252.146, and 266.159 through the use of preparative HPLC. NMR data allowed for the identification of the metabolites as: Lansamide I (**19**) [11], Lansiumamide B (**20**) [12], Lansiumamide C (**21**) [12], and *N*-(2-phenylethyl)cinnamamide (**22**) [13] (Figure 2). In the chromatogram shown in Figure 1B, the characteristic peaks of these compounds are highlighted in red. Noteworthy, compounds (**19**) and (**20**) are configurational isomers, whose *m*/*z* is 264.104. Furthermore, compounds represented by *m*/*z* 280.144 and 282.156 (Figure 3) are correlated with metabolites found in another Rutaceae plant, *Clausena lansium* [14] as well as the isolated and identified compounds.

The GNPS platform was important to identify compound (**26**), whose *m*/*z* is 307.186, as (*E*)-*N*-(4-acetamidobutyl)-3-(4-hydroxy-3-methoxyphenyl)prop-2-enamide. These data confirm the consistent result for grouping the compounds in cluster I, which is also highlighted by the obtained cosine values (higher than 0.7), pointing to significant fragmentation similarities among the clustered compounds. The comparison between the experimental and registered (GNPS database) spectra (Figure 4) also demonstrates the resemblances around the fragmentation pattern, which was important for compound identification.

Molecular family II is basically formed by acridones, a class of natural products quite characteristic in *Swinglea glutinosa* [4,5,15]. In this work, some of them were isolated and identified: citrusinine-I (**1**) [16], citrusinine (**2**) [17], glycotrycine IV (**5**) [18], and 5-hydroxynoracronycine (**6**) [19]. In addition, the presence of cluster II also suggests the likely production of other alkaloids that have not been reported for *S. glutinosa* yet. The nodes represented by *m*/*z* 312.091, *m*/*z* 370.134, and *m*/*z* 318.102 did not show any correlation with our “in-house” database. The last one was identified using MS/MS spectra comparison at the GNPS platform as 1,3,6-trihydroxy-4,5-dimethoxy-10-methylacridin-9-one (**23**) [20]. Therefore, our approach revealed the potential of finding untapped acridones in *S. glutinosa*.

In its turn, for molecular family III, it was observed as a flavonoid cluster, some of whose compounds were identified according to MS/MS spectra matches through the GNPS database [2]. The candidates suggested for *m*/*z* 565.164, *m*/*z* 579.179, and *m*/*z* 549.169 were 5,7-dihydroxy-2-(4-hydroxyphenyl)-8-[3,4,5-trihydroxy-6-(hydroxymethyl)oxanitrile)-2-yl]-6-(3,4,5-trihydroxyoxan-2-yl)chromen-4-one (**27**), 5-hydroxy-7-[3,4,5-trihydroxy-6-(hydroxymethyl)oxan-2-yl]-oxy-2-[4-(3,4,5-trihydroxy-6-methoxoxan-2-yl)oxiphenyl]chromen-4-one (**28**), and 5,7-dihydroxy-2-phenyl-6-[3,4,5-trihydroxy-6-(hydroxymethyl)oxan-2-yl]-8-(3,4,5-trihydroxyoxan-2-yl)chromen-4-one (**29**), respectively. Furthermore, clusters IV–VIII were also observed, but any corresponding metabolite was identified using the described analytical tools.

Employing the two mentioned dereplication strategies, it was possible to identify 29 compounds, 11 of them not described for the *Swinglea* genus. These methodologies guided the isolation of four phenylacrylamides, alkaloid-based compounds that were also first shown in the plant genus.

In a nutshell, the use of the combined approaches has been useful for exploring the chemical profile of the *Swinglea* genus, in particular regarding the detection of alkaloid-based compounds produced by the plant. Altogether, the results point toward still hidden specialized metabolites from *Swinglea glutinosa* to be revealed in the ongoing work.

## 3. Materials and Methods

### 3.1. Target Analysis and Molecular MS/MS Networking-Based Dereplication

A list creation for target candidates in the Target Analysis 1.3 (Bruker Daltonics^®^, Bremen, Germany) program processing was performed through the Microsoft Excel interface, with the compound name and the molecular formula, according to the literature information. Considered processing parameters were SigmaFit at 1000 (broad, isotope-free), 60 (medium), 20 (low), mass accuracy accessed lower than 5 ppm, and mSigma lower than 50. Area cut-off was set to 2000 counts as the default and DataAnalysis 4.2 software (Bruker Daltonics^®^) was used for manual comparison of extracted-ion chromatograms (EIC) generated by Target Analysis.

For MS/MS dereplication via molecular networking analysis (GNPS), MS/MS data were acquired using AutoMS mode and converted to .mzXML format using MS-Convert software, which is part of ProteoWizard (Palo Alto, CA, USA). The networks were generated using the online platform (https://gnps.ucsd.edu/ProteoSAFe/static/gnps-splash.jsp) [2]. All MS/MS peaks within ±17 Da deviations from the precursor ions were filtered out. MS/MS spectra were selected from only the six best peaks, considering a range of ±50 Da across the spectrum. The data were grouped with a tolerance of 0.02 Da for precursor ions and 0.02 Da for fragment ions in the construction of “consensus” spectra (identical spectra for each precursor, which are combined to create the node to be visualized). Consensus spectra with less than two spectra were not considered. Connections between nodes were filtered to values greater than 0.7 of the cosine parameter, with compatibility for more than six peaks. For the dereplication of compounds, the generated network spectra were consulted at the GNPS libraries, using the same selection criteria for the analyzed samples. GNPS data were analyzed and viewed using Cytoscape 3.7.0 software (U.S. National Institute of General Medical Sciences, Bethesda, MD, USA).

### 3.2. Acridone Alkaloids and Phenylacrylamides Isolation and Identification

The plant material was divided into two parts, stem and leaves, followed by drying in an air circulation oven at 40 °C. After grinding, materials were submitted to extraction by maceration in ethanol for three days. After three days, the ethanol was filtered off and evaporated. The procedure was repeated until the third extraction to obtain the extracts from the stems and leaves of *S. glutinosa*. In sequence, from the ethanolic crude extracts, the liquid–liquid extraction procedure was employed to prepare hexane, ethyl acetate, and butanol fractions.

The stems hexane fraction (0.93 g) was subjected to silica column chromatography (diameter: 4.0 cm; height: 1 cm) using hexane, ethyl acetate, and methanol as the gradient mode eluent yielding 16 subfractions (A1 to A16). Fractions A6 and A7 were submitted to preparative HPLC (C12—Synergi Max column—150 mm × 4.60 mm, 4 μ), allowing the isolation of one substance from A6 (6) and three substances from A7 (1, 2, and 5). The employed mobile phase was formed by acetonitrile (ACN) and H_2_O (both with the addition of 0.1% formic acid) and the method used for all these substances was: 0.01–2.5 min—15% ACN; 2.5–12 min—15–95% ACN; 12–20 min—95% ACN; 20–23 min—95–15% ACN; 23–27 min—15% ACN; this procedure allowed us to obtain citrusinine-I (1) (5.0 mg) [16], citibrasine (2) (11.7 mg) [17], glycotrycine IV (**5**) (21.2 mg) [18], and 5-hydroxynoracronycine (**6**) (2.0 mg) [19].

The hexane fraction from the leaves (8.0 g) of *S. glutinosa* was fractionated using a silica chromatography column (diameter: 5.7 cm; height: 30 cm); hexane, ethyl acetate, and methanol were used as gradient mode eluents yielding 10 fractions (B1–B10). Fractions B5 and B7 were submitted to preparative HPLC (column C18—250 mm × 4.6 mm—Luna 5 μ). The mobile phase used was ACN and H_2_O (both with addition of 0.1% formic acid) and the method used for isolation was: 0.01–2.5 min—60% ACN; 2.5–12 min—60–95% ACN; 12–20 min—95% ACN; 20–23 min—95–60% ACN; 23–27 min—60% ACN; this procedure allowed us to obtain Lansamide I (**19**) (8.1 mg) [11], Lansiumamide B (**20**) (3.8 mg) [12], Lansiumamide C (**21**) (20.6 mg) [12], and *N*-(2-phenylethyl)cinnamamide (**22**) (18.0 mg) [13]. The NMR spectra were recorded on a Bruker spectrometer (Bruker Daltonics^®^) Ultrashield 300—Advance III operating at 300 MHz (^1^H) and 75 MHz (^13^C). The spectra are presented in the Appendix A.

Lansamide I (**19**): ^1^H NMR (CDCl_3_, δ (*J*/Hz)): 7.77 (d, 16.4); 7.32 (d, 14,0); 7.20–7.59 (m); 7.02 (d, 15.4); 6.07 (d, 14.3); 3.37 (s); ^13^C NMR (CDCl_3_): 135.2; 130.2; 129.1; 128.2; 128.9; 126.8; 125.8; 117.3; 29.8. HRMS. *m*/*z* 264.1379 [M + H]^+^ (calcd for C_18_H_17_NO, Δ3.4 ppm).

Lansiumamide B (**20**): ^1^H NMR (CDCl_3_, δ (*J*/Hz)): 7.55 (d, 15.0); 7.20–7.34 (m); 6.93 (d, 15.0); 6.50 (d, 8.6); 6.24 (d, 8.7); 3.09 (s). ^13^C NMR (CDCl_3_): 141.4; 135.4; 124.1; 127.5–129.5; 118.9; 33.6. HRMS. *m*/*z* 264.1381 [M + H]^+^ (calcd for C_18_H_17_NO, Δ2.6 ppm).

Lansiumamide C (**21**): ^1^H NMR (CDCl_3_, δ (*J*/Hz)): 7.72 (d, 15.4); 7.20–7.41 (m); 6.58 (d, 15.4); 3.71 (q, 7.2); 3.07 (s); 2.94 (t, 7.4). ^13^C NMR (CDCl_3_): 166.5; 142.2; 140.6–127.0; 119.5; 51.9; 36.1; 34.3. HRMS. *m*/*z* 266.1554 [M + H]^+^ (calcd for C_18_H_19_NO, Δ3.3 ppm).

*N*-(2-phenylethyl)cinnamamide (**22**): ^1^H NMR (CDCl_3_): 7.54 (d, 15.5); 7.22–7.50 (m); 6.31 (d, 15.6); 5.60 (s); 3.67 (q, 6.5); 2.89 (t, 7.0). ^13^C NMR (CDCl_3_): 140.5; 130.2–127.0; 123.0; 41.6; 36.5. HRMS. *m*/*z* 252.1397 [M + H]^+^ (calcd for C_17_H_17_NO, Δ3.5 ppm).

## Figures and Tables

**Figure 1 molecules-25-00153-f001:**
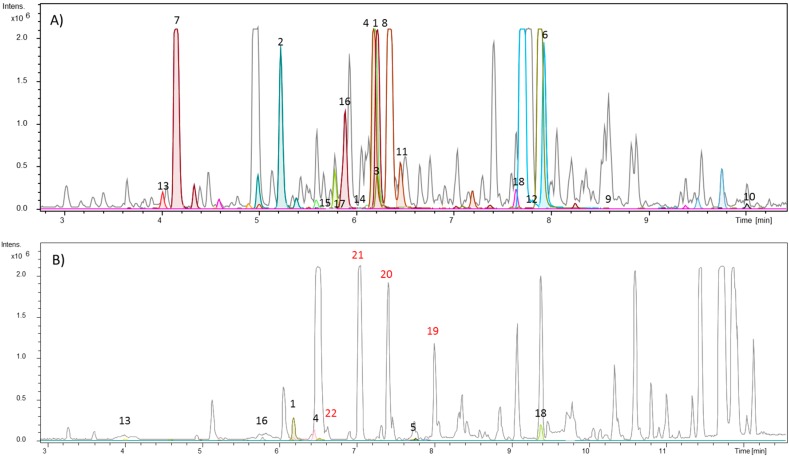
(**A**) Base peak chromatogram (BPC) of *S. glutinosa* hexane stem fractions. (**B**) BPC of *S. glutinosa* hexane leaf fraction. The chromatogram is overlaid with the extracted-ion chromatogram from detected compounds. The colored peaks represent compounds listed in the “in-house” database, some of them identified in Table 1 and Figure 2. The peaks numbered in red correspond to the isolated amides in this work, not yet reported for the genus.

**Figure 2 molecules-25-00153-f002:**
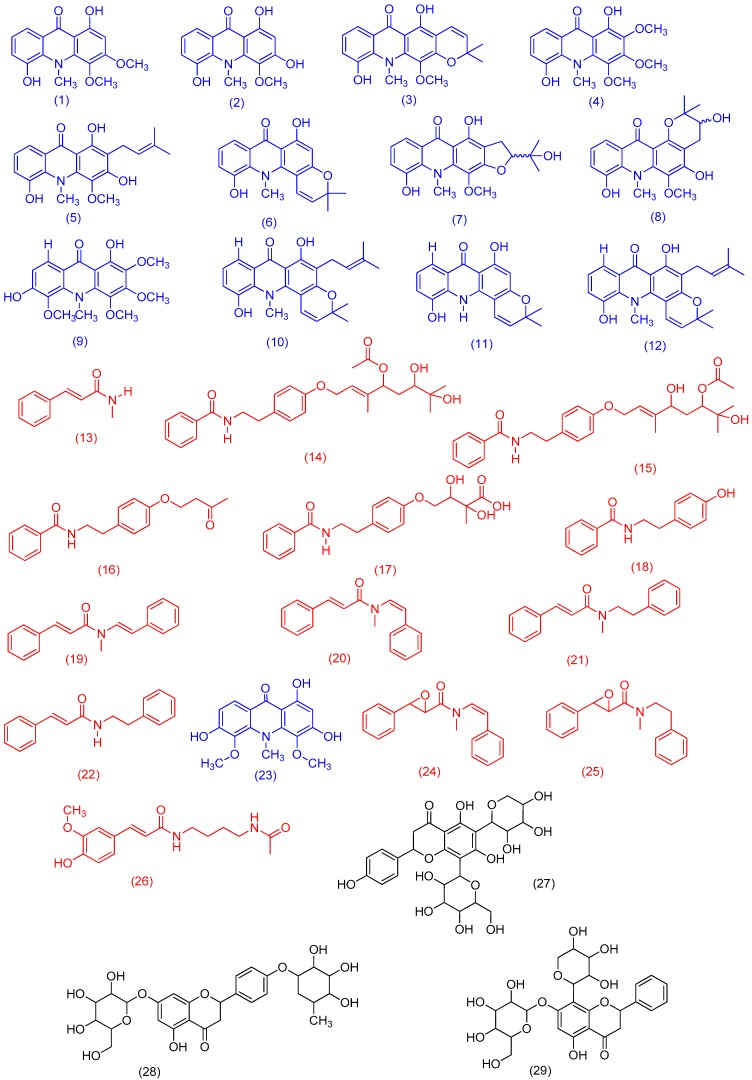
Identified compounds from *Swinglea glutinosa* through UHPLC-HRMS (compounds **1**–**22**, using Target Analysis; compounds **23**–**29** using GNPS). The compounds indicated in red correspond to the phenylacrylamide class; the compounds in blue belong to the acridonic alkaloid class and in black are compounds belonging to the flavonoid class.

**Figure 3 molecules-25-00153-f003:**
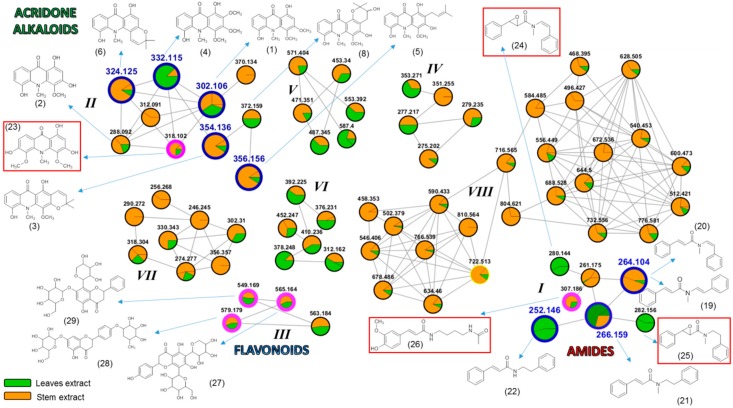
Molecular families for *S. glutinosa* extracts. Nodes outlined in blue represent isolated and identified compounds in this work. The nodes outlined in pink represent dereplicated compounds, which had the chemical structure suggested by the GNPS platform. Compounds indicated from non-prominent nodes suggest substances compatible with metabolites already described for *S. glutinosa*. Structures highlighted in the red frame indicate compounds not related to the *Swinglea* genus and that were identified by our “in-house” database. Different portions visualized at nodes are not quantitatively representative.

**Figure 4 molecules-25-00153-f004:**
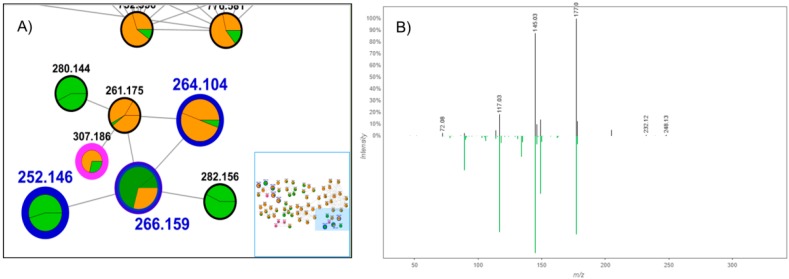
(**A**) Part of molecular family I, corresponding to amide detection, with highlighted cosine values. (**B**) MS/MS spectrum acquired (black) vs. registered spectrum on the GNPS platform (green), for the ion *m*/*z* 307.186. The pseudomolecular ion was not detected in both cases.

**Table 1 molecules-25-00153-t001:** Identified compounds from the hexane stem and hexane leaf fractions of *Swinglea glutinosa* through UHPLC-HRMS (using Target Analysis), their molecular formulas, exact masses, and accurate masses.

Compound Name (Code)	Molecular Formula	Exact Mass	Accurate Mass [M + H]^+^
citrusinine-I (**1**)	C_16_H_15_NO_5_	301.0950	302.1018
citrusinine-II (**2**)	C_15_H_13_NO_5_	287.0794	288.0862
pyranofoline (**3**)	C_20_H_19_NO_5_	353.1263	354.1334
citibrasine (**4**)	C_17_H_17_NO_6_	331.1056	332.1131
glycotrycine IV (**5**)	C_20_H_21_NO_5_	355.1420	356.1488
5-hydroxynoracronycine (**6**)	C_19_H_17_NO_4_	323.1158	324.1230
2,3-dihydro-4,9-dihydroxy-2-(2-hydroxypropan-2-yl)-11-methoxy-10-methylfuro[3,2-b]acridin-5(10*H*)-oneI (**7**)	C_20_H_21_NO_6_	371.1369	372.1439
3,4-dihydro-3,5,8-trihydroxy-6-methoxy-2,2,7-trimethyl-2*H*-pyrano[2,3-a]acridin-12(7*H*)-one (**8**)	C_20_H_21_NO_6_	371.1369	372.1452
des-*N*-methylnoracronycine (**9**)	C_19_H_17_NO_3_	307.1208	308.1250
5-hydroxy-*N*-methylseverifoline (**10**)	C_24_H_25_NO_4_	391.1784	392.1842
glyfoline (**11**)	C_18_H_19_NO_7_	361.1162	362.1272
atalaphyllinine (**12**)	C_23_H_23_NO	377.1627	378.1446
(*E*)-*N*-methylcinnamamide [(*E*)-*N*-methylphenylacrylamide] (**13**)	C_10_H_11_NO	161.0841	162.0912
*N*-benzoyl-*O*-(4-acetoxyl-6,7-dihydroxy)geranylthiramine (**14**)	C_27_H_35_NO_6_	469.2464	470.2522
*N*-benzoyl-*O*-(6-acetoxyl-4,7-dihydroxy)geranylthiramine (**15**)	C_27_H_35_NO_6_	469.2464	470.2522
*N*-{2-[4-(butoxy-3-one) phenyl]ethylbenzamide (**16**)	C_19_H_21_NO_3_	311.1521	312.1591
*N*-{2 [4-(2,3-dihydroxy-2-methylbutoxyethyl)phenyl] ethylbenzamide (**17**)	C_20_H_23_NO_6_	373.1525	374.1573
*N*-benzoyltyramine (**18**)	C_15_H_15_NO_2_	241.1103	242.1175
lansamide I (**19**)	C_18_H_17_NO	263.131	264.1379
lansiumamide B (**20**)	C_18_H_17_NO	263.131	264.1381
lansiumamide C (**21**)	C_18_H_19_NO	265.147	266.1554
*N*-(2-phenylethyl)cinnamamide (**22**)	C_17_H_17_NO	251.131	252.1397

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
