# Peer review of "Integrated Analytical Tools for Accessing Acridones and Unrelated Phenylacrylamides from Swinglea glutinosa"

_molecules, 2019, doi:10.3390/molecules25010153_

Round 1
Reviewer 1 Report
This work described an integrated analytical tools applied to natural products to avoid the purification of known metabolites. The strategies imply two step: target analysis software (Bruker daltonics) and the GNPS plataform. Based on this plataform the authors isolated from Swinglea glutinosa four alkaloids reported for the first time: Lansamide I, Lansiumamide B, 27 Lansiumamide C and N-(2-phenylethyl)-cinnamamide.
I think that this MS is well done. I strongly suggest the authors can explain the ethylacetate and butanol extracts were not studied. I am not native English speaker but I think that all manuscript need to improve English level. It also suggests the authors to check the manuscript since in this way presents many mistakes.
Please, the author should describe the HPLC/MS data in more details.
Please, the author should check the all references since exists mistakes for instance ref 12.
Please, check molecular cluster III. The authors should include trivial name for all flavonoids.
Please, include in Supp. Inf. fragmentation scheme for all compounds detected.
Author Response
Following the suggestions of all referees, we sent the manuscript for language review. The review was conducted by the Cambridge proofreding editors. Besides that, the main changes are highlighted in yellow directly on the file.
Referee:I strongly suggest the authors can explain the ethylacetate and butanol extracts were not studied
Ethylacetate has already been studied by our group. Please take a quick look at Ferreira et al. (2016) Chem. Biodiversity, 13, 100 – 106. The butanol extract is being studied by the PhD student Ana Calheiros de Carvalho (first author of this manuscript).
Referee: Please, the author should check the all references since exists mistakes for instance ref 12.
We have checked the reference number 12. Thanks for your suggestion.
Referee: Please, check molecular cluster III. The authors should include trivial name for all flavonoids.
Thanks for your suggestion, but we did not find trivial names for the flavonoids.
Referee: Please, the author should describe the HPLC/MS data in more details. Please, include in Supp. Inf. fragmentation scheme for all compounds detected.
We really appreciate your suggestion, but we believe that fragmentation schemes are necessary only for the isolated amides in this manuscript (compounds 19 – 22). Thus, we prepared fragmentation schemes for them. Please, take a look at figures A15 – A18 (Supp. Material).
Thanks for all yours suggestions.
Reviewer 2 Report
The manuscript entitled “Integrated analytical tools for accessing acridones and unrelated phenylacrylamides from Swinglea glutinosa” reports an interesting approach which can be applied for the preliminary screening of complex natural products mixtures e for well-addressing fractionation and isolation of pure compounds in phytochemical research.
The authors well organize and present data. Well done!
Author Response
Following the suggestions of the three referees, we sent the manuscript for language review. The review was conducted by the Cambridge proofreding editors. Besides that, the main changes are highlighted in yellow directly on the file.
Thanks for your positive feedback.
Reviewer 3 Report
The authors present integrated analytical tools for the chemical study of Swinglea glutinosa. A combination of Target analysis MS data analysis and MS/MS data from GNPS platform was used to verify the identity of previously characterized components of the extract and led to the characterization of several new compounds.
In the introduction, lines 50-66, a stronger case should be made for why the authors chose to study Swinglea glutinosa. There is generic mention to the class of acridones present in the plant to have desirable biological activity, but the description is general and lacks detail. What components from the extract have activity? Is there potential for their use as a drug? What is the novelty or potential for meaningful results to come from this work?
The caption to Figure 1 does not well address the content presented in the figure. In Figure 1A, there appears to be at least three overlapping chromatograms. The caption refers to two chromatograms. The peaks could use to be labeled to correspond to the compounds identified in Figure 2 and Table 1. It is difficult to determine the main point the authors seek to convey with this figure. The point should be explicitly stated and the figure should correlate well with the point.
The connection between Figure 1 and the information provided in Figure 2 and Table 1 is not made well. Perhaps a labeling of peaks in Figure 1 chromatograms with numbers that correlate to the molecules identified in Figure 2 and characterized in Table 1 would link the figures together.
For readers not familiar with the data representation provided in Figure 3, it is necessary to indicate what the roman numerals represent. What is the difference between molecular families and clusters? Neither term is really defined. Please indicate what they represent.
On page 7, line 127, should the reference be to Figure 2 instead of Figure 1?
Page 7, lines 131-135, a combination of molecular family and cluster is used. Neither term has been defined. Please clarify the similarity or difference of the titles.
Figure 4, is there any relationship between molecular family 1 and the compounds listed in Figure 2? If the families and clusters are important, why not organize the content in Figure 2 to appear according to molecular family and/or cluster?
What was the significance of the new molecules characterized in this study? Are they new? Have they been characterized prior? How do the results advanced the field?
Author Response
First of all, thanks for your fair and critical feedback!
Following the suggestions of the three referees, we sent the manuscript for language review. The review was conducted by the Cambridge proofreding editors. Besides that, the main changes are highlighted in yellow directly on the file.
Referee: In the introduction, lines 50-66, a stronger case should be made for why the authors chose to study Swinglea glutinosa. There is generic mention to the class of acridones present in the plant to have desirable biological activity, but the description is general and lacks detail. What components from the extract have activity? Is there potential for their use as a drug? What is the novelty or potential for meaningful results to come from this work?
Thanks for your observations. Actually, we are continuing our studies with Swinglea glutinosa (please check reference number 5). First publication was in 2016, and we just found acridones, which were characterized as inhibitors of photosynthesis. At that point we did not applied any dereplication approaches. Now, we decided to explore the plan metabolism by modern tools to discovery unrelated compounds. Our approach allowed us to discovery 11 substances not reported yet to Swinglea genus. Besides that, in the new version we reported with more details biological activities associated to acridones; please take a look at lines 90-106.
As said reviewer 2 our manuscript “reports an interesting approach which can be applied for the preliminary screening of complex natural products mixtures e for well-addressing fractionation and isolation of pure compounds in phytochemical research”. Thus, I guess we are presenting novelties in the field.
Referee: The caption to Figure 1 does not well address the content presented in the figure. In Figure 1A, there appears to be at least three overlapping chromatograms. The caption refers to two chromatograms. The peaks could use to be labeled to correspond to the compounds identified in Figure 2 and Table 1. It is difficult to determine the main point the authors seek to convey with this figure. The point should be explicitly stated and the figure should correlate well with the point.
In Figure 1, there are only two overlapping chromatograms. One of them is the base peak chromatogram (BPC), obtained from the crude fraction. The second one is the extract chromatogram ion (EIC) from the compounds listed in our in-house database. The colored bands represent compounds which are on both chromatogram, BPC and EIC.
Referee: The connection between Figure 1 and the information provided in Figure 2 and Table 1 is not made well. Perhaps a labeling of peaks in Figure 1 chromatograms with numbers that correlate to the molecules identified in Figure 2 and characterized in Table 1 would link the figures together.
To facilitate the comprehension, we labeled the compounds numbers on the chromatograms, as you suggested. Thank you so much.
Referee: For readers not familiar with the data representation provided in Figure 3, it is necessary to indicate what the roman numerals represent. What is the difference between molecular families and clusters? Neither term is really defined. Please indicate what they represent.
The roman numerals represent an attempt to identify the observed cluster; is a "kind" of code to accelerate the text reading. Besides that, in this manuscript we tried to use CLUSTER and MOLECULAR FAMILY as synonyms. However, to avoid confusion, in this new version we changed everything to cluster (s).
Referee: On page 7, line 127, should the reference be to Figure 2 instead of Figure 1?
Thanks. We promoted the change.
Referee: Page 7, lines 131-135, a combination of molecular family and cluster is used. Neither term has been defined. Please clarify the similarity or difference of the titles.
In the new version, we decided to standardize all for CLUSTERS.
Referee: Figure 4, is there any relationship between molecular family 1 and the compounds listed in Figure 2? If the families and clusters are important, why not organize the content in Figure 2 to appear according to molecular family and/or cluster?
In the new version, we organized Figure 2 according to the class of compounds, differentiating each class with different colors.
Referee: What was the significance of the new molecules characterized in this study? Are they new? Have they been characterized prior? How do the results advanced the field?
As said reviewer 2 our manuscript “reports an interesting approach which can be applied for the preliminary screening of complex natural products mixtures e for well-addressing fractionation and isolation of pure compounds in phytochemical research”. Thus, I guess the questions were kindly answered by him.
Round 2
Reviewer 3 Report
The authors have satisfactorily addressed the reviewer comments.